# Fair Personalized Learner Modeling Without Sensitive Attributes

## Abstract

Personalized learner modeling uses learners' historical behavior data to diagnose their cognitive abilities, a process known as Cognitive Diagnosis (CD) in the literature. This is a fundamental yet crucial task in web-based learning services, such as learning resource recommendation and adaptive testing. Previously, researchers discovered that models improperly correlate learners' abilities with their sensitive attributes, resulting in unfair diagnoses for learners from different sensitive groups (e.g., gender, region). Given the input of sensitive attributes, researchers proposed decorrelating these attributes from the modeling process, demonstrating improved fairness results. However, privacy concerns make collecting sensitive attributes impractical. This challenge is compounded by the presence of multiple sensitive attributes, making fairness improvement under any of them difficult. In this paper, we explore how to achieve fair personalized learner modeling without relying on any sensitive attribute input. Specifically, we first introduce a novel fairness objective tailored for personalized learner modeling. We then propose a max-min strategy that facilitates both potential sensitive information inference and fair CD modeling. In the max step, we propose a pseudo-label inference method based on maximizing the designed fairness objective. Given these pseudo-labels, the min step involves retraining a fair CD model by minimizing the designed objective. Additionally, we provide a theoretical guarantee that implementing our proposed framework reduces the upper bound of fairness generalization error. Extensive experiments demonstrate that the proposed framework significantly outperforms existing methods in terms of fairness and accuracy. Our code is available at https://anonymous.4open.science/r/FairWISA-40C6/.

## Keywords

Fairness, User Modeling, Cognitive Diagnosis

## 1 Introduction

In recent years, online learning platforms such as Coursera[1] and ASSISTments[2] have rapidly emerged, offering personalized web learning services [1] like exercise recommendations [2, 3] and adaptive testing [4]. In these services, personalized learner modeling [5] plays a crucial role, focusing on capturing learners' cognitive states through their online behavioral data. Among the various techniques employed in personalized learner modeling, CD [6–9] has gained widespread adoption. Through comprehensive modeling of learners, CD provides diagnostic feedback to both platforms and learners, enabling informed decisions about learning paths and performance improvements in personalized web learning services.

In recent research endeavors, improving the accuracy of CD models has been the central theme [6, 10, 13–17]. Despite considerable progress in accuracy, it has been reported that existing CD models unconsciously introduce unfairness [18, 19]. Here, unfairness refers to situations where models show prejudice or favoritism toward

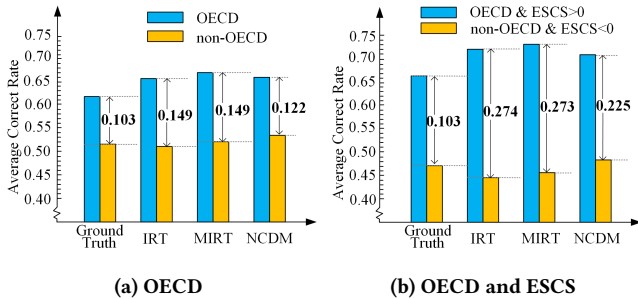

**(a) OECD**       **(b) OECD and ESCS**

**Figure 1: Unfairness in CD. Average Correct Rate is the average percentage of correct answers within each group. Bolding numbers denote the average correct rate gaps between groups. IRT [10], MIRT [11], and NCDM [12] are currently popular CD models, respectively. OECD and ESCS[4] are two sensitive attributes.**

particular learner groups based on their sensitive attributes (e.g., gender, region). *This unfairness typically manifests as CD models widening the proficiency gap between different learner groups.* Fig. 1 (a) illustrates the existence of unfairness in three popular CD models on PISA[3] dataset. Specifically, we compare the predicted proficiency levels of learners from OECD and non-OECD countries[4]. It can be observed that these CD models consistently amplify the proficiency gap between OECD and non-OECD learners, with the predicted correct rate gap exceeding the actual correct rate gap between these groups. This occurs because learner records inherently correlate with sensitive attributes. When optimizing for accuracy, CD models may unintentionally learn these correlations, resulting in unfair predictions [20–22]. Fairness is crucial to ensure equitable treatment and prevent discrimination against specific learner groups [23]. Therefore, developing fair CD models is of paramount importance.

Algorithmic fairness has gained significant attention in recent research, with most existing methods relying on specific sensitive attribute labels [22, 24–26]. However, the CD task presents some special challenges. *First, obtaining learners' sensitive attributes is often impractical.* Privacy concerns and legal restrictions (such as the GDPR[5] in the European Union and FERPA[6] in the USA) strictly regulate the collection and processing of students' personal data [27, 28]. *Second, numerous hidden attributes in online learning may lead to unfairness.* Unfortunately, most existing research only addresses unfairness caused by a single known attribute, failing to cover all types of unfairness in CD. We find that multiple sensitive attributes can exacerbate unfairness, so we need a method to

---

[1]https://www.coursera.org/
[2]https://new.assistments.org/

[3]PISA dataset is an international education dataset that is widely used in research on intelligent education and is described in detail in Appendix C.1.
[4]OECD and ESCS are universally acknowledged as sensitive attributes in educational research and assessment in PISA. Details are provided in Appendix C.1.
[5]The General Data Protection Regulation, a comprehensive data protection law enacted by the European Union in 2018. For more information, visit *https://gdpr.eu/.*
[6]The Family Educational Rights and Privacy Act, a federal law on the protection of learner privacy enacted in the United States in 1974. For more information, visit *https://www2.ed.gov/policy/gen/guid/fpco/ferpa/.*

eliminate the impact caused by all of them. As shown in Fig. 1(b), the gap in predicted correct rates between OECD learners from advantaged families (ESCS>0) and non-OECD learners from disadvantaged families (ESCS<0) is further amplified, compared to the single-attribute scenario in Fig. 1(a). The two unique characteristics mentioned above prompt us to develop an advanced fair CD model that doesn't rely on sensitive attributes.

In this paper, *we investigate the challenging yet practical research task of improving the fairness of CD models without relying on sensitive attributes.* The absence of labeled sensitive attributes creates significant obstacles to achieving fair CD modeling: (1) *Lack of Fairness Supervision Signals.* Without the labeled sensitive attributes, guiding the model towards fairness is difficult. Some methods attempt to infer these labels using limited sensitive information [19, 29] or associated non-sensitive features [30], but acquiring such substitutes introduces new challenges. (2) *Multiple Potentially Sensitive Attributes.* Fair CD emphasizes equitable performance across all sensitive groups. Consequently, methods optimizing only for specific groups are unsuitable for this task [31, 32]. (3) *Theoretical Guarantee.* Establishing a theoretical guarantee for fair learner modeling becomes problematic when using data with unavailable sensitive attributes. (4) *Framework Compatibility.* A general fair framework is needed to mitigate unfairness for existing CD models.

To bridge these gaps, we propose a model-agnostic fair CD framework that does not rely on any sensitive attribute information, named **Fair** cognitive diagnosis **WI**thout **S**ensitive **A**ttributes (*Fair-WISA*). Specifically, we design a novel fairness objective function for measuring unfairness levels in CD modeling. Then, we propose a max-min training game for both sensitive attribute inference and unfairness mitigation. More concretely, we propose a pseudo-label inference method based on maximizing the designed fairness objective. These inferred pseudo-labels are used as proxies for sensitive information in fair CD modeling. Given the pseudo-labels, we retrain a fair CD model by minimizing the designed fairness objective across every group. Our proposed optimization method ensures fairness across all sensitive attributes, not just specific ones. We also provide a theoretical guarantee that the implementation of *FairWISA* is equivalent to reducing the upper bound of the error optimization function for fairness generalization. In the experiments, we evaluate our proposed framework under both real-world and out-of-distribution settings, with the latter presenting more challenging conditions for CD tasks. Extensive experimental results on these two settings clearly show the effectiveness of our proposed framework. In summary, the primary contributions of this paper are as follows:

- We introduce a challenging yet practical research problem of improving CD models' fairness without relying on any sensitive attribute information. To address this, we propose a model-agnostic fair framework.

- We design a novel fairness objective function to optimize the CD model's fairness. We invent a max-min training method for both sensitive attribute inference and unfairness mitigation. Furthermore, we give a theoretical guarantee for our proposed framework.

- Extensive experiments on real-world datasets and challenging datasets have been conducted to validate the effectiveness of our proposed framework.

## 2 Related Works and Preliminaries

### 2.1 Cognitive Diagnosis

Cognitive diagnosis [13, 14] focuses on assessing learners' proficiency levels based on historical learner-exercise interaction logs, which are important for intelligent education and web learning [33, 34]. Let $U = \{u_1, u_2, ..., u_N\}$ and $E = \{e_1, e_2, ..., e_M\}$ be the sets of learners and exercises, respectively. The learner-exercise interaction set is denoted as $R = \{(u, e, y_{ue})|u \in U, e \in E, y_{ue} \in \{0, 1\}\}$, where $y_{ue}$ indicates whether $u$ answers exercise $e$ correctly. Specifically, if $u$ answers $e$ correctly, then $y_{ue} = 1$; otherwise, $y_{ue} = 0$. The inputs to the CD model consist of the learner-exercise interaction records $R$. The CD model analyzes these records to assess the learner's mastery of each exercise, thereby inferring the learner's overall proficiency level. Generally, the CD model contains two steps: 1) the embedding layers to obtain the diagnostic factors of learners and exercises, 2) the interaction layer to learn the interaction function $f(\cdot)$ among the factors and output the probability $\hat{y}_{ue}$ of learner $u$ correctly answering exercise $e$.

$$\hat{y}_{ue} = f(h_u, h_e), \tag{1}$$

where $h_u$ is the proficiency vector of $u$ and $h_e$ is the exercise vector. Both the embedding layer architecture and interaction function can be flexibly designed. Many CD methods [6, 10–12, 35, 36] have been proposed to model learners' abilities more accurately. For example, Item Response Theory (IRT) [10, 11] models learners' abilities using one-dimensional vectors and employs a logit function to depict the interactions between learners and exercises. NCDM [6, 12] employs multidimensional vectors to represent learners, with each dimension reflecting the student's mastery of specific knowledge, and utilizes a neural network to capture the complex interactions between students and exercises.

When training the CD model, for each record in the response logs set $R$, the loss function $\mathcal{L}_{CD}$ is calculated as the cross-entropy loss between the predicted value $\hat{y}_{ue}$ and the true label $y_{ue}$:

$$\mathcal{L}_{CD} = - \sum_{(u,e,y_{ue}) \in R} (y_{ue} \log \hat{y}_{ue} + (1 - y_{ue}) \log(1 - \hat{y}_{ue})). \tag{2}$$

After training, the CD model obtains learners' proficiency levels.

However, most existing work focuses only on improving the accuracy of cognitive diagnosis, ignoring the fairness issues that exist in the modeling process. Recently, some scholars have begun to focus on studying fair cognitive diagnosis [18, 19]. Our work differs from these studies by focusing on a more challenging scenario in which we do not use sensitive labels as supervisory signals.

### 2.2 Fairness in Cognitive Diagnosis

Fairness aims to ensure that models do not exhibit bias or discrimination when processing data from users with different sensitive attributes[37] such as gender and ethnicity. Existing studies on fairness can generally be divided into two categories [38]: individual fairness [20, 39] and group fairness [18, 40, 41]. Individual fairness stipulates that similar individuals should be treated similarly, while group fairness focuses on ensuring equitable outcomes across different sensitive groups. Group fairness has garnered more research attention due to its clearer definitions and measurements. For example, Demographic Parity (*DP*) [20] requires that different groups have equal correct rates. However, it is limited as the base

correct rates of subgroups differed. Equal Opportunity (*EO*) [42] emphasizes the need for true positive rates (TPR) across different groups. It ensures that people with equivalent abilities have equal opportunities to achieve positive outcomes, regardless of their sensitive attributes. Recent developments in cognitive diagnosis have introduced new perspectives on fairness, emphasizing that cognitive diagnostic models should operate independently of learners' sensitive attributes [18, 19]. These studies propose specific criteria for fairness in CD models. Zhang et al. [18] argue that fair CD models should not change the original correct rate gaps between sensitive groups. Zhang et al. [19] focus on the fairness of learners' representations in CD, asserting that these representations should not reveal learners' sensitive attributes. Building on these insights, this paper focuses on group fairness in cognitive diagnosis.

### 2.3 Fairness-aware Models

In tackling fairness concerns, many studies [43–46] have been proposed to confront the bias inherent in historical data. These works largely address the issue of fairness in machine learning and can generally be categorized into three types: pre-processing methods [24, 47], in-processing methods [45, 48], and post-processing methods [42, 49]. Pre-processing methods typically alleviate bias in data by correcting labels [47], modifying sensitive attributes [50], and generating balanced samples [24]. Most studies assume that sensitive attributes are fully available. However, in reality, learner privacy protection and technical constraints often make these attributes unavailable. Recently, researchers have focused on improving fairness without access to sensitive attributes. To address this challenge, some studies [19, 29] use optimal discriminators to predict these attributes. Other studies [30–32, 51] enhance task-specific fairness without sensitive attributes. Lahoti et al. [31] proposed adversarial reweighting learning to achieve Rawlsian maximum-minimum fairness. Zhao et al. [30] achieved fairness by minimizing the correlations between predictions and non-sensitive features that are similar to sensitive features. Chai et al. [32] used knowledge distillation to achieve fairness without sensitive attributes by modifying the training data labels to soft labels, which is equivalent to weighting the special sample.

Some other fairness-aware methods [52] related to our work focus on improving model generalization when group information is unknown. For example, [53] and [54] propose methods that learn to weight samples, assigning higher weights to complex samples to improve robustness. Creager et al. [55] inferred environment labels by maximally violating the Environment Invariance Constraint, and then use the inferred labels for invariant learning to get causal representations. [56] and [57] obtain environment labels by clustering samples and then co-optimise with the invariant learning task to enhance generalization. These efforts aim to enhance the model's performance beyond its original distribution and are not specifically designed to address fairness bias.

## 3 The Proposed *FairWISA* Framework
### 3.1 Overview

The core of *FairWISA* is a max-min training game for both sensitive group inference and unfairness mitigation. We begin by designing a fairness objective to quantify the unfairness in CD models. Based on this objective, *FairWISA* first maximizes the degree of unfairness

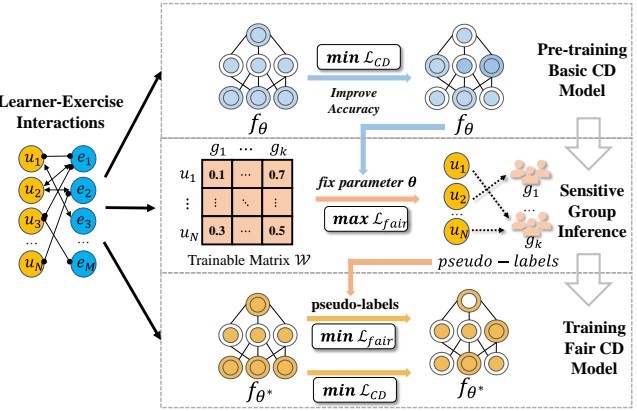

**Figure 2: The framework of *FairWISA*.**

to infer pseudo-labels for learners' group membership. Then, it minimizes the gaps between different groups to achieve fair CD modeling. As illustrated in Fig. 2, *FairWISA* comprises three main procedures:

**1) Pre-training a basic CD Model**: This initial step involves developing an unfair CD model by maximizing accuracy. As previously noted, this accuracy-centric approach inherently introduces bias. The basic CD model can be selected from various established models such as IRT [10], MIRT [11], or NCDM [12], which are detailed in Appendix A.

**2) Sensitive Group Inference**: This phase involves grouping learners by fixing the parameters of the model from Step 1) and then maximizing the degree of unfairness to derive pseudo-sensitive labels for each learner. This approach is effective because the pre-trained model from Step 1) exhibits differential treatment towards groups with distinct sensitive attributes. Therefore, significant performance disparities between groups indicate differing sensitive attributes. We leverage this characteristic to infer pseudo-labels as proxies for learners' sensitive attributes.

**3) Training the Fair CD Model**: In this final step, we utilize the pseudo-labels obtained from Step 2) as fairness supervision signals to guide the training of CD. To achieve a balance between accuracy and fairness in the CD model, we employ a regularized optimization approach that simultaneously addresses both objectives.

In the subsequent sections, we first formulate the fairness objective and then introduce our sensitive group inference method. Following this, we provide a detailed description of the *FairWISA* training process and present a theoretical analysis of its plausibility.

### 3.2 Fairness Objective $\mathcal{L}_{fair}$

Previous fairness objectives, such as *EO* [42] and $F_{CD}$ [18], are based on predicted outcomes, which can be significantly influenced by classification thresholds. Therefore, we propose a threshold-free fairness optimization objective based on predicted values. This objective is motivated by the key observation: there is a correlation between the CD model's predicted values and learners' sensitive attributes.

**Observation.** We counted the predicted values of the CD model for both OECD and non-OECD learners in the PISA dataset and plotted the distribution of these predicted values as shown in Fig. 3.

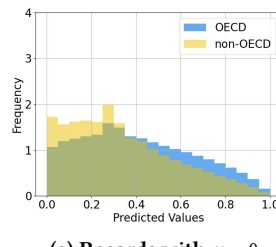 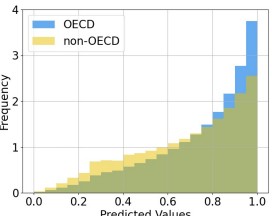

**(a) Records with $y = 0$**    **(b) Records with $y = 1$**

**Figure 3: CD model's predicted value distributions of different sensitive groups. The left panel shows the distribution of the CD model's predicted value $\hat{y}$ for records with incorrectness ($y = 0$), and the right for correctness ($y = 1$). The model consistently predicts higher values for the OECD group.**

As can be seen from the figure, the distribution of predicted values for OECD learners (blue area) is consistently to the right of the distribution of predicted values for non-OECD learners (yellow area), despite both groups having the same answer results. Specifically, the distribution of predicted values for OECD learners trends closer to 1, while the distribution for non-OECD learners trends closer to 0. This suggests that the CD model consistently outputs larger predicted values for OECD learners, regardless of whether the answers are actually incorrect (Fig. 3(a)) or correct (Fig. 3(b)).

Based on the observation presented above, we argue that the discrepancies in the model's predicted values across different learners result in unfair diagnoses. To address this, we design a novel fairness objective named $\mathcal{L}_{fair}$, which regularizes the model-predicted values $\hat{y}_{ue}$ across different groups. $\mathcal{L}_{fair}$ ensures that the CD model outputs similar predicted values for learners with similar proficiency levels, regardless of sensitive attributes.

$$\mathcal{L}_{fair} = Var(\mathcal{R}^0_{g_1}, ..., \mathcal{R}^0_{g_k}) + Var(\mathcal{R}^1_{g_1}, ..., \mathcal{R}^1_{g_k}), \quad (3)$$

$$\mathcal{R}^0_{g_i} = \frac{1}{N^0_{g_i}} \sum_{u \in g_i, y_{ue}=0} \hat{y}_{ue}, \quad (4)$$

$$\mathcal{R}^1_{g_i} = \frac{1}{N^1_{g_i}} \sum_{u \in g_i, y_{ue}=1} \hat{y}_{ue}, \quad (5)$$

where $Var(\cdot)$ denotes the variance calculation operator, $\mathcal{R}^0_{g_i}$ and $\mathcal{R}^1_{g_i}$ are calculated by Eq. (4) and Eq. (5), respectively. $k$ is the number of sensitive attribute groups, $N^0_{g_i}(N^1_{g_i})$ is the number of records with incorrect (correct) responses in the group $g_i$. $\mathcal{L}_{fair}$ brings closer the average values of the model's outputs across different sensitive groups when the true label is 0 or 1, respectively. For Eq. (4), when $Var(\mathcal{R}^0_{g_1}, ..., \mathcal{R}^0_{g_k}) = 0$, indicating $\mathcal{R}^0_{g_1} = \mathcal{R}^0_{g_2} = ... = \mathcal{R}^0_{g_k}$, it can be inferred that the CD model treats different groups comparably and is therefore fair. The same analysis applies to Eq. (5). Therefore, a larger $\mathcal{L}_{fair}$ indicates greater unfairness with values ranging from 0 to 1. It is worth noting that although $\mathcal{L}_{fair}$ is proposed for CD, it can still be applied to other classification tasks.

Compared to some previous fairness metrics (e.g., $DP$ [20], $EO$ [42]), $\mathcal{L}_{fair}$ has several advantages as a fairness optimization objective: (1) Previous metrics required discretizing predicted values, a process that is generally non-differentiable for optimization and may be affected by classification threshold (e.g., whether $\hat{y} > 0.5$

or $\hat{y} > 0.6$ is regarded as a correct answer may yield significant differences). In contrast, $\mathcal{L}_{fair}$ is based on original predicted values and thus does not suffer significantly from such interference. (2) Considering that in CD tasks, both correct and incorrect answers reflect the proficiency level of the learner, $\mathcal{L}_{fair}$ focuses on both types of answers (as shown in Eq. (4) and Eq. (5)), which is not addressed by other existing fairness metrics. This suggests that $\mathcal{L}_{fair}$ is better adapted to the two-sided fairness of answering correctly or incorrectly, whereas other metrics only consider the one-sided. Therefore, $\mathcal{L}_{fair}$ is more suitable for fair CD modeling.

### 3.3 Sensitive Group Inference

CD models perform unfairly when processing samples from different sensitive demographic groups, as evidenced by previous studies [18, 19]. Leveraging this characteristic, we have devised a novel grouping method. The core concept involves fixing the parameters of a pre-trained CD model and using unfairness maximization as a supervisory signal to group learners. The rationale behind this approach is as follows: Pre-trained CD models tend to evaluate learners similarly within sensitive attribute groups, but differently across these groups. This leads to assessment discrepancies between different sensitive groups. Larger discrepancies indicate higher levels of model unfairness. It is worth noting that our proposed $\mathcal{L}_{fair}$ serves as an effective proxy for measuring this discrepancy. When we fix the model parameters and group learners, a larger assessment gap between two groups suggests a greater dissimilarity in the sensitive attributes of learners within these groups. Based on this idea, we achieve learner grouping by maximizing the degree of unfairness (as quantified by the $\mathcal{L}_{fair}$).

Following the idea of maximizing the $\mathcal{L}_{fair}$ to identify sensitive groups, we devise a grouping approach as follows. Before inference, a pre-trained CD model that has learned the inherent biases and treats learners from different sensitive groups differently is required. The grouping matrix $\mathcal{W} \in \mathbb{R}^{N \times k}$ is learned to record the group information of all learners, where $N$ is the number of learners and $k$ is a hyperparameter representing the number of groups. Each row of $\mathcal{W}$ contains the learner's group probabilities, and the sum of the elements in each row of $\mathcal{W}$ is 1, i.e.,

$$\mathcal{W}_{u1} + \mathcal{W}_{u2} + ... + \mathcal{W}_{uk} = 1, \quad (6)$$

where $\mathcal{W}_{ug}$ denotes the probability that learner $u$ belongs to group $g$. Then, the group label of $u$ is determined by the column in row $u$ of $\mathcal{W}$ where the maximum value is located. Specifically, the group label of $u$ is given by:

$$\arg\max_g \mathcal{W}_{ug}. \quad (7)$$

Initially, the elements in $\mathcal{W}$ are randomly initialized, meaning learners' groups are assigned randomly. Then, to assign learners with similar sensitive attributes to the same group as much as possible, we make maximizing $\mathcal{L}_{fair}$ as the supervisory signal, thereby updating $\mathcal{W}$, i.e.,

$$\arg\max_{\mathcal{W}} \mathcal{L}_{fair}. \quad (8)$$

Given the pseudo group label $g_i$, we can feed it to Eq. (4) and Eq. (5) to calculate $\mathcal{L}_{fair}$. Therefore, Eq. (8) transforms into:

$$\mathcal{W} = \arg\max_{\mathcal{W}} Var(\mathcal{R}^0_{g_1}, ..., \mathcal{R}^0_{g_k}) + Var(\mathcal{R}^1_{g_1}, ..., \mathcal{R}^1_{g_k}). \quad (9)$$

After training, the group labels of each learner can be obtained from $\mathcal{W}$.

As described above, frequent operations of selecting the maximum value are required to obtain the group labels for the learners during during the training of $\mathcal{W}$. However, the operation of selecting the maximum value is non-differentiable. Therefore, we employ the Gumbel-Softmax [58] function to handle this issue, ensuring that training gradients can be propagated. Gumbel-Softmax [58] is a technique used in deep learning for handling discrete choices. The process is formulated as:

$$\mathcal{W}'_{ug} = \frac{exp\left(\frac{log(\mathcal{W}_{ug})+gb}{\tau}\right)}{\sum_{g'} exp\left(\frac{log(\mathcal{W}_{ug'})+gb}{\tau}\right)}, \tag{10}$$

where $\mathcal{W}_{ug}$ denote the probability that the learner $u$ belongs to group $g$, $\mathcal{W}'_{ug}$ denotes the converted output, $gb$ is a sample from the Gumbel distribution, and $\tau$ is the temperature parameter controlling the level of smoothing. This approach enables the training of neural networks by making the discrete selection process differentiable.

### 3.4 Training Process

*FairWISA* executes the following three steps. **1)** Firstly, we pre-train an unfair basic CD model $\mathcal{F}_\theta$ by Eq. (2). Note that the choice of basic model is arbitrary. **2)** Then, the sensitive group inference module is executed to get sensitive group labels. Specifically, we first initialize the grouping matrix $\mathcal{W}$ randomly. Subsequently, keeping the parameters of the pre-trained CD model $\mathcal{F}_\theta$ fixed, we maximize $\mathcal{L}_{fair}$ to infer sensitive group labels. After training, an optimized grouping matrix $\mathcal{W}$ is obtained, which can be used to get the pseudo-labels of the learners. **3)** Finally, the fair CD model is trained using the pseudo-labels obtained from the well-learned $\mathcal{W}$. Specifically, we fix the $\mathcal{W}$ obtained in the previous step and optimize a new CD model by minimizing $\mathcal{L}_{total}$, which incorporates $\mathcal{L}_{fair}$ as a regularization term. This optimization yields the fair CD model $\mathcal{F}_{\theta^*}$. The total loss function is defined as:

$$\mathcal{L}_{total} = \mathcal{L}_{CD} + \alpha \mathcal{L}_{fair}, \tag{11}$$

where $\alpha$ is the hyperparameter of fairness regularization. The pseudocode for the implementation is in Appendix B.

By leveraging $\mathcal{L}_{fair}$ as the fairness objective and employing the group label inference process, *FairWISA* is capable of enhancing fairness with the inferred group labels as proxies.

### 3.5 Theoretical Analysis

In CD model training, sensitive attributes are mistakenly associated with the output, leading the model to learn shortcuts rather than the learner's actual proficiency level. This often occurs due to biased sampling or labeling in the training data. Similarly, generalization issues occur when models rely on shortcuts, resulting in poor performance on new data. Therefore, recent studies have shown that fairness problems are related to out-of-distribution issues [59]. From this perspective, improving model fairness can be viewed as enhancing generalization under specific distributional shifts. Thus, the fairness problem can be viewed as a special case of the generalization problem. The goal of fair CD is to develop models that do not rely on associations between sensitive attributes and predicted outcomes, performing well across unknown target distributions. To this end, we conduct analyses from the perspective

of out-of-distribution generalization to demonstrate that *FairWISA* is theoretically supported. Our analysis demonstrates that implementing *FairWISA* effectively reduces the upper bound of the error optimization function for fairness generalization, thereby proving its effectiveness. Drawing inspiration from prior research [60, 61], we have the following proposition.

**Proposition 1.** *(Proposition 2.1 in [60]) Let $\mathcal{X}$ be a space, $\mathcal{H}$ be a class of hypotheses corresponding to this space, and $d_{\mathcal{H}\Delta\mathcal{H}}$ be the $\mathcal{H}$-divergence that measures distributional differences. Let $\mathbb{Q}$ be the target distribution and the collection $\{\mathbb{P}_i\}_{i=1}^k$ be distributions over $\mathcal{X}$ and let $\{\varphi_i\}_{i=1}^k$ be a collection of non-negative coefficients with $\sum_i \varphi_i = 1$. Let $O$ be a set of distributions such that for every $\mathbb{S} \in O$ the following holds:*

$$\sum_i \varphi_i d_{\mathcal{H}\Delta\mathcal{H}}(\mathbb{P}_i, \mathbb{S}) \leq \max_{i,j} d_{\mathcal{H}\Delta\mathcal{H}}(\mathbb{P}_i, \mathbb{P}_j). \tag{12}$$

*Then, for any $h \in \mathcal{H}$, the error on the target domain $\mathbb{Q}$, denoted as $\varepsilon_{\mathbb{Q}}(h)$, is proven to satisfy the following [60]:*

$$\varepsilon_{\mathbb{Q}}(h) \leq \underbrace{\lambda_\varphi}_{I} + \underbrace{\sum_i \varphi_i \varepsilon_{\mathbb{P}_i}(h)}_{II} + \underbrace{\frac{1}{2}\min_{\mathbb{S}\in O} d_{\mathcal{H}\Delta\mathcal{H}}(\mathbb{S}, \mathbb{Q})}_{III} + \underbrace{\frac{1}{2}\max_{i,j} d_{\mathcal{H}\Delta\mathcal{H}}(\mathbb{P}_i, \mathbb{P}_j)}_{IV}, \tag{13}$$

*where $\lambda_\varphi = \sum_i \varphi_i \lambda_i$ and each $\lambda_i$ is the error of an ideal joint hypothesis for $\mathbb{Q}$ and $\mathbb{P}_i$, $\varepsilon_{\mathbb{P}_i}(h)$ is the error for a hypothesis $h$ on a distribution $\mathbb{P}_i$.*

From Proposition 1, the upper bound of the model's error in the unseen target domain $\mathbb{Q}$ can be expressed as Eq. (13). A lower value of $\varepsilon_{\mathbb{Q}}(h)$ indicates better generalization performance of the model. Then, we analyze each term of Eq. (13).

For term $I$, $\lambda_\varphi$ can be ignored in practice because it is small in reality. For term $II$, $\sum_i \varphi_i \varepsilon_{\mathbb{P}_i}(h)$ represents the error in the training domain. Empirical Risk Minimization (ERM) is an appropriate method for controlling this term. *FairWISA* optimizes it by minimizing $\mathcal{L}_{CD}$. For term $III$, $\frac{1}{2}\min_{\mathbb{S}\in O} d_{\mathcal{H}\Delta\mathcal{H}}(\mathbb{S}, \mathbb{Q})$ is the smallest $\mathcal{H}$-divergence between $\mathbb{S}$ and $\mathbb{Q}$. Given that $\mathbb{Q}$ is unknown, the only way to reduce this term is to expand the range of $O$, thereby increasing the likelihood of finding an $\mathbb{S}$ that is closer to $\mathbb{Q}$. According to Eq. (12), maximizing the distribution gap between $\mathbb{P}_i$ and $\mathbb{P}_j$ achieves this. In *FairWISA*, we infer group labels by maximizing $\mathcal{L}_{fair}$, which increases the distributional disparity between groups. For term $IV$, $\frac{1}{2}\max_{i,j} d_{\mathcal{H}\Delta\mathcal{H}}(\mathbb{P}_i, \mathbb{P}_j)$ represents the maximum pairwise $\mathcal{H}$-divergence among the source domains. *FairWISA* minimizes $\mathcal{L}_{fair}$ to reduce the differences between different domains, thereby decreasing the value of $\frac{1}{2}\max_{i,j} d_{\mathcal{H}\Delta\mathcal{H}}(\mathbb{P}_i, \mathbb{P}_j)$.

The above analysis provides theoretical insights into the steps *FairWISA* takes: minimizing $\mathcal{L}_{CD}$ for improving accuracy, maximizing $\mathcal{L}_{fair}$ to amplify the distribution gap between groups for inferring sensitive group labels, and minimizing $\mathcal{L}_{fair}$ for improving fairness.

# 4 Experiments

## 4.1 Datasets and Experiment Settings

**Datasets.** We conduct experiments on two widely used education datasets: PISA [7] and SLP [8]. The detailed descriptions of PISA and SLP are provided in Appendix C.1.

**Evaluation metrics.** In the experiments, we evaluate both the accuracy and group fairness of the CD models using appropriate metrics, allowing for comparison against baseline methods. For accuracy, the Area Under the ROC Curve (AUC) and Accuracy (ACC) are used, with higher values indicating better accuracy. For fairness, $EO$ [42], $NEO$ [62], and $F_{CD}$ [18] are employed, where values closer to 0 reflect more equitable treatment across different groups. The detailed descriptions of the fairness metrics are as follows:

$$EO = |P(\hat{y} = 1 \mid y = 1, s = 0) - P(\hat{y} = 1 \mid y = 1, s = 1)|, \quad (14)$$

$$NEO = |P(\hat{y} = 0 \mid y = 0, s = 0) - P(\hat{y} = 0 \mid y = 0, s = 1)|, \quad (15)$$

where $s$ is the sensitive attribute, $y$ is the label, and $\hat{y}$ is the model's prediction. In cognitive diagnosis, $EO$ (or $NEO$) reflects the disparity in the proportion of correct predictions among records of correct (or incorrect) answers across different groups.

$$F_{CD} = \left( \frac{1}{|A|} \sum_{i \in A} \hat{Y}_i - \frac{1}{|B|} \sum_{i \in B} \hat{Y}_i \right) - \left( \frac{1}{|A|} \sum_{i \in A} Y_i - \frac{1}{|B|} \sum_{i \in B} Y_i \right), \quad (16)$$

where $|A|(|B|)$ is the number of learners in group $A(B)$, $Y$ is the actual correct rate of learner, and $\hat{Y}$ is the predicted correct rate. The closer $F_{CD}$ is to 0, the less unfair bias the CD model amplifies.

It has been shown that many studies have demonstrated an inherent trade-off between model accuracy and fairness [24, 30, 32, 48]. As this paper focuses on fairness modeling, the accuracy metrics across models are controlled to be comparable, enabling clearer insights into the relative fairness performance achieved.

**Baselines.** We conduct comparisons with several baselines, which are provided as follows:

- **Original model**. It serves as the basic CD model without additional fairness guarantees. In the experiment, three CD models are employed: IRT [10], MIRT [11], and NCDM [6, 12].
- **F_Attr**. It first assumes that the sensitive attribute *Attr* is available and divides learners into groups based on *Attr*. Then, it uses $\mathcal{L}_{total}$ (Eq. (11)) as the loss function to optimize the CD models.
- **Reg_EO**. It first assigns group labels to learners as per the group inference method in Section 3.3, and then uses $EO$ [42] as the fairness regularization for training.
- **ARL** [31]. It is a method based on Rawlsian fairness principle of Max-Min welfare. It optimises by reweighting samples through adversarial learning.
- **KD** [32]. This approach replaces hard labels with soft labels during training, using knowledge distillation techniques. By doing so, it achieves fairness through effectively weighting samples equally. Specifically, we use an overfitting trained CD model as the "Teacher" model in the method, and use the predicted values of the "Teacher" model as the soft labels to train the "Student" model.

---

[7]https://www.oecd.org/pisa/data/

[8]https://aic-fe.bnu.edu.cn/en/data/index.html

- **EIIL** [55]. It is a method of invariant learning that does not rely on group labels, inferring environmental labels by maximizing the violation of the invariance principle, and then using invariant learning to acquire invariant representations, thereby enhancing fairness.

**Parameter settings.** *FairWISA* and all baseline methods are implemented using the PyTorch library. For model initialization, all parameters are set to a Gaussian distribution with a mean of 0 and a standard variance of 0.01. As the optimizer, Adam is used with a learning rate of 0.001 for the IRT-based and MIRT-based methods, and a learning rate of 0.0005 for the NCDM-based methods. The batch size is 4096 for the PISA dataset and 2048 for the SLP dataset. Additionally, the dimensions for both learners and exercises are set to 8 for the MIRT-based methods.

## 4.2 Experimental Results

**Performance on accuracy and fairness (RQ1).** We compare *Fair-WISA* with baselines on the three CD models as backbones. Table 1 and Table 2 show the accuracy and fairness performance on multiple sensitive attributes on the PISA and SLP datasets. The results indicate that all original CD models exhibit unfairness, underscoring the imperative to investigate fairness in cognitive diagnosis. And For all the methods, as fairness is enhanced, concomitant reductions in accuracy are observed. This is a typical fairness-accuracy trade-off pattern.

As evidenced in both Table 1 and Table 2, *FairWISA* substantially enhances the fairness while accuracy losses are constrained within 2.5% relative to backbones (IRT, MIRT, NCDM). This suggests that *FairWISA* is effective in modeling CD fairness without using sensitive information. And compared to baselines, *FairWISA* consistently demonstrated lower values for $EO$, $NEO$, and $F_{CD}$ in the majority of cases, while maintaining closely comparable accuracy metrics. This suggests that *FairWISA* is more competitive in terms of fairness performance. It is observed that there are some anomalies in the results presented in the tables. For example, in the SLP dataset, *FairWISA* performs slightly worse compared to the baselines when NCDM is used as the backbone. This may be due to the higher number of parameters for NCDM and the lower amount of data in SLP, and thus the unlearned sufficient unfairness bias for getting guidance information of grouping.

In summary, *FairWISA* proves effective in enhancing the fairness of cognitive diagnosis without significant compromises in accuracy and it is competitive with baselines.

**Performance on Robustness (RQ2).** Here, we introduce a "Challenging Test" scenario to evaluate performance under group distributional biases. This test creates a fairness distribution that differs from the training data by varying the correlation between sensitive attribute groups and labels. Details of the design process are in Appendix C.3. The motivation behind this design is to comprehensively assess *FairWISA*'s adaptability across diverse fairness distributions.

Table 3 and Table 4 show the results of the experiments on PISA and SLP, respectively. Compared to Tables 1 and 2, all methods in Tables 3 and 4 exhibited a decrease in accuracy (AUC and ACC). This is because the CD models tend to learn the unfair bias of the data during training to improve the accuracy under the same distribution. Thus, despite data distribution shifts, the CD models still

**Table 1: Comparison of different approaches on PISA. *OECD*, *Gender*, and *ESCS* serve as sensitive attributes, dividing data samples into two groups for fairness evaluation. Bolding signifies the best performance among all methods.**

| | | IRT | ARL | EIIL | KD | FairWISA | MIRT | ARL | EIIL | KD | FairWISA | NCDM | ARL | EIIL | KD | FairWISA |
|---|---|---|---|---|---|---|---|---|---|---|---|---|---|---|---|---|
| AUC ↑ | | 0.8049 | 0.7833 | 0.7809 | 0.7820 | 0.7827 | 0.7995 | 0.7802 | 0.7838 | 0.7909 | 0.7813 | 0.7771 | 0.7713 | 0.7744 | 0.7666 | 0.7749 |
| ACC ↑ | | 0.7316 | 0.7134 | 0.7138 | 0.7119 | 0.7130 | 0.7273 | 0.7117 | 0.7146 | 0.7202 | 0.7106 | 0.7135 | 0.7108 | 0.7064 | 0.7089 | 0.7101 |
| EO ↓ | OECD | 0.0970 | 0.0791 | 0.0847 | 0.0789 | **0.0491** | 0.0931 | 0.0665 | 0.0744 | 0.0828 | **0.0437** | 0.0737 | 0.0656 | 0.0644 | 0.0706 | **0.0591** |
| | Gender | 0.0435 | 0.0360 | 0.0376 | 0.0367 | **0.0262** | 0.0419 | 0.0323 | 0.0354 | 0.0379 | **0.0246** | 0.0327 | 0.0294 | 0.0290 | 0.0316 | **0.0283** |
| | ESCS | 0.1122 | 0.0926 | 0.0959 | 0.0936 | **0.0597** | 0.1062 | 0.0779 | 0.0826 | 0.0952 | **0.0529** | 0.0866 | 0.0777 | 0.0774 | 0.0846 | **0.0690** |
| NEO ↓ | OECD | 0.1197 | 0.0969 | 0.1088 | 0.0948 | **0.0633** | 0.1204 | 0.0854 | 0.0923 | 0.1068 | **0.0566** | 0.0898 | 0.0759 | 0.0761 | 0.0794 | **0.0706** |
| | Gender | 0.0597 | 0.0473 | 0.0539 | 0.0468 | **0.0351** | 0.0619 | 0.0437 | 0.0476 | 0.0535 | **0.0337** | 0.0435 | 0.0368 | 0.0386 | 0.0373 | **0.0368** |
| | ESCS | 0.1111 | 0.0864 | 0.1017 | 0.0862 | **0.0508** | 0.1091 | 0.0723 | 0.0795 | 0.0943 | **0.0420** | 0.0788 | 0.0662 | 0.0601 | 0.0693 | **0.0581** |
| $F_{CD \to 0}$ | OECD | 0.0498 | 0.0243 | 0.0338 | 0.0232 | **-0.0071** | 0.0460 | **0.0110** | 0.0210 | 0.0321 | -0.0138 | 0.0182 | 0.0070 | 0.0033 | 0.0116 | **0.0006** |
| | Gender | 0.0235 | 0.0113 | 0.0154 | 0.0109 | **-0.0003** | 0.0225 | 0.0071 | 0.0118 | 0.0155 | **-0.0013** | 0.0076 | 0.0025 | 0.0030 | 0.0041 | **0.0015** |
| | ESCS | 0.0477 | 0.0208 | 0.0305 | 0.0211 | **-0.0130** | 0.0416 | **0.0054** | 0.0137 | 0.0272 | -0.0216 | 0.0143 | **0.0035** | 0.0070 | 0.0087 | -0.0052 |

**Table 2: Comparison of different approaches on SLP. *Gender* and *Income* serve as sensitive attributes, dividing data samples into two groups for fairness evaluation. Bolding signifies the best performance among all methods.**

| | | IRT | ARL | EIIL | KD | FairWISA | MIRT | ARL | EIIL | KD | FairWISA | NCDM | ARL | EIIL | KD | FairWISA |
|---|---|---|---|---|---|---|---|---|---|---|---|---|---|---|---|---|
| AUC ↑ | | 0.8806 | 0.8652 | 0.8496 | 0.8650 | 0.8675 | 0.8777 | 0.8604 | 0.8594 | 0.8593 | 0.8606 | 0.8439 | 0.8399 | 0.8401 | 0.8396 | 0.8420 |
| ACC ↑ | | 0.8350 | 0.8295 | 0.8119 | 0.8267 | 0.8256 | 0.8337 | 0.8229 | 0.8230 | 0.8229 | 0.8213 | 0.8153 | 0.8129 | 0.8127 | 0.8101 | 0.8112 |
| EO ↓ | Gender | 0.0580 | 0.0454 | 0.0532 | 0.0409 | **0.0396** | 0.0541 | 0.0519 | 0.0433 | 0.0553 | **0.0409** | 0.0564 | 0.0315 | 0.0341 | **0.0308** | 0.0476 |
| | Income | 0.0417 | 0.0336 | 0.0308 | 0.0321 | **0.0299** | 0.0405 | 0.0367 | 0.0344 | 0.0385 | **0.0327** | 0.0432 | 0.0228 | 0.0270 | **0.0222** | 0.0377 |
| NEO ↓ | Gender | 0.1124 | 0.0842 | 0.0981 | 0.0734 | **0.0572** | 0.1211 | 0.1329 | 0.0723 | 0.1260 | **0.0527** | 0.0783 | **0.0225** | 0.0296 | 0.0239 | 0.0520 |
| | Income | 0.0713 | 0.0611 | 0.0556 | 0.0470 | **0.0378** | 0.0731 | 0.0649 | 0.0524 | 0.0693 | **0.0426** | 0.0527 | 0.0217 | 0.0267 | **0.0151** | 0.0381 |
| $F_{CD \to 0}$ | Gender | 0.0241 | **0.0025** | 0.0164 | -0.0053 | -0.0058 | 0.0222 | 0.0171 | -0.0092 | 0.0185 | **-0.0053** | 0.0110 | -0.0268 | -0.0220 | -0.0287 | **-0.0014** |
| | Income | 0.0146 | **0.0016** | 0.0051 | -0.0060 | -0.0058 | 0.0125 | **0.0026** | -0.0035 | 0.0070 | -0.0033 | 0.0068 | -0.0217 | -0.0156 | -0.0249 | **-0.0011** |

**Table 3: Comparison of different approaches on PISA challenging test. *OECD*, *Gender*, and *ESCS* serve as sensitive attributes, dividing data samples into two groups for fairness evaluation. Bolding signifies the best performance among all methods.**

| | | IRT | ARL | EIIL | KD | FairWISA | MIRT | ARL | EIIL | KD | FairWISA | NCDM | ARL | EIIL | KD | FairWISA |
|---|---|---|---|---|---|---|---|---|---|---|---|---|---|---|---|---|
| AUC ↑ | | 0.7081 | 0.7065 | 0.7013 | 0.7073 | **0.7448** | 0.7059 | 0.7112 | 0.7155 | 0.7075 | **0.7338** | 0.7101 | 0.7204 | 0.7215 | 0.7181 | **0.7220** |
| ACC ↑ | | 0.6531 | 0.6521 | 0.6464 | 0.6524 | **0.6854** | 0.6533 | 0.6596 | 0.6571 | 0.6549 | **0.6761** | 0.6571 | 0.6615 | 0.6597 | 0.6564 | **0.6655** |
| EO ↓ | OECD | 0.0556 | 0.0436 | 0.0491 | 0.0422 | **0.0133** | 0.0549 | 0.0373 | 0.0443 | 0.048 | **0.0232** | 0.0419 | 0.0361 | 0.0355 | 0.039 | **0.0331** |
| | Gender | 0.0252 | 0.0217 | 0.0238 | 0.0213 | **0.0008** | 0.0238 | 0.0158 | 0.0165 | 0.0211 | **0.0045** | 0.0199 | 0.0173 | 0.0173 | 0.0198 | **0.0131** |
| | ESCS | 0.0736 | 0.0610 | 0.0645 | 0.0619 | **0.0219** | 0.0704 | 0.0534 | 0.0546 | 0.0639 | **0.0337** | 0.0582 | 0.0523 | 0.0538 | 0.0577 | **0.0464** |
| NEO ↓ | OECD | 0.0342 | 0.0285 | 0.0315 | 0.0293 | **0.0159** | 0.0346 | 0.0264 | 0.0261 | 0.032 | **0.0138** | 0.0266 | 0.0231 | 0.0209 | 0.0253 | **0.0208** |
| | Gender | 0.0019 | 0.0005 | 0.0010 | **0.0001** | 0.0042 | 0.0032 | 0.0048 | 0.0080 | **0.0031** | 0.0067 | **0.0002** | 0.0003 | 0.0044 | 0.0010 | 0.0046 |
| | ESCS | 0.0539 | 0.0394 | 0.0429 | 0.0397 | **0.0200** | 0.0509 | 0.0271 | 0.0332 | 0.0405 | **0.0120** | 0.0361 | 0.0296 | 0.0249 | 0.0312 | **0.0211** |
| $F_{CD \to 0}$ | OECD | 0.5286 | 0.5057 | 0.5244 | 0.5044 | **0.4060** | 0.5282 | 0.4876 | 0.4946 | 0.5126 | **0.4377** | 0.4933 | 0.4737 | 0.4725 | 0.4815 | **0.4677** |
| | Gender | 0.3645 | 0.3539 | 0.3642 | 0.3530 | **0.3050** | 0.3655 | 0.3468 | 0.3503 | 0.3586 | **0.3228** | 0.3468 | 0.3364 | 0.3386 | 0.3392 | **0.3355** |
| | ESCS | 0.5268 | 0.5011 | 0.5223 | 0.5006 | **0.3948** | 0.5240 | 0.4803 | 0.4867 | 0.5067 | **0.4273** | 0.4887 | 0.4685 | 0.4594 | 0.4764 | **0.4588** |

**Table 4: Comparison of different approaches on SLP challenging test. *Gender* and *Income* serve as sensitive attributes, dividing data samples into two groups for fairness evaluation. Bolding signifies the best performance among all methods.**

| | | IRT | ARL | EIIL | KD | FairWISA | MIRT | ARL | EIIL | KD | FairWISA | NCDM | ARL | EIIL | KD | FairWISA |
|---|---|---|---|---|---|---|---|---|---|---|---|---|---|---|---|---|
| AUC ↑ | | 0.8141 | 0.8017 | 0.8009 | 0.8023 | **0.8211** | 0.8088 | 0.7864 | 0.7864 | 0.7826 | **0.8098** | 0.7822 | 0.7994 | 0.7958 | 0.7930 | 0.7915 |
| ACC ↑ | | 0.7228 | 0.7136 | 0.7234 | 0.7096 | **0.7263** | 0.7169 | 0.6940 | 0.6940 | 0.6945 | **0.7198** | 0.7097 | 0.6952 | 0.7101 | 0.7002 | **0.7173** |
| EO ↓ | Gender | 0.0957 | 0.0773 | 0.0855 | 0.0705 | **0.0655** | 0.0918 | 0.0863 | 0.0863 | 0.0893 | **0.0735** | 0.1026 | **0.0389** | 0.0634 | 0.0546 | 0.0876 |
| | Income | 0.0762 | 0.0607 | 0.0621 | 0.0562 | **0.0557** | 0.0787 | 0.0743 | 0.0743 | 0.0710 | **0.0605** | 0.0775 | **0.0307** | 0.0496 | 0.0403 | 0.0637 |
| NEO ↓ | Gender | 0.1089 | 0.0774 | 0.0870 | 0.0699 | **0.0525** | 0.1125 | 0.1149 | 0.1149 | 0.1117 | **0.0461** | 0.0674 | **0.0002** | 0.0222 | 0.0121 | 0.0441 |
| | Income | 0.0652 | 0.0402 | 0.0554 | 0.0404 | **0.0215** | 0.0637 | 0.0625 | 0.0625 | 0.0839 | **0.0352** | 0.0623 | **0.0104** | 0.0171 | 0.0214 | 0.0292 |
| $F_{CD \to 0}$ | Gender | 0.1315 | 0.1114 | 0.1190 | 0.1065 | **0.0917** | 0.1345 | 0.1367 | 0.0966 | 0.1335 | **0.0905** | 0.1139 | **0.0537** | 0.0709 | 0.0664 | 0.0918 |
| | Income | 0.5681 | 0.5640 | 0.5267 | 0.5582 | **0.5018** | 0.5783 | 0.6135 | 0.0542 | 0.6178 | **0.5184** | 0.5661 | 0.5207 | 0.5199 | 0.5220 | **0.5179** |

use learned unfair bias for predictions, resulting in reduced accuracy when the data is out of distribution. It is also observed that *FairWISA* exhibits the highest accuracy metrics (AUC and ACC) among all the baselines. It indicates that *FairWISA* is more adaptive when faced with this fairness distribution shift; on the other hand, it also suggests that *FairWISA* enables fairer cognitive diagnosis. Additionally, *FairWISA* performs competitively across all fairness metrics (*EO*, *NEO*, $F_{CD}$) on various sensitive attributes when compared to all the baselines. In conclusion, *FairWISA* exhibits strong robustness when the distribution of fairness bias is shifted.

### 4.3 Discussion

**Improving fairness of all sensitive attributes.** We use F_*Attr* for comparison to discuss the necessity of the study without sensitive attributes. Specifically, for each dataset, we divide the learners into two groups according to the known sensitive attributes and observe the fairness performance of F_*Attr*. Taking "Gender" as the known sensitive attribute, F_*Gender* divides learners into two groups: "male" and "female" and utilizes $\mathcal{L}_{total}$ (Eq. (11)) as the optimization objective. Note that F_*Attr* utilizes known sensitive attributes, while *FairWISA* does not.

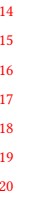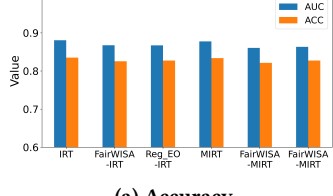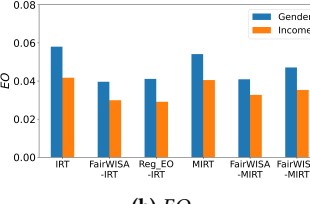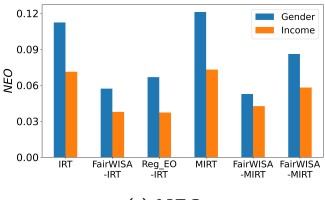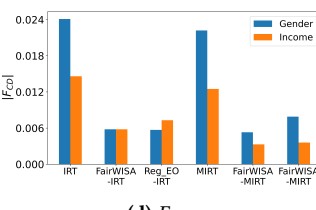

|  |  |  |  |
|---|---|---|---|
| (a) Accuracy | (b) $EO$ | (c) $NEO$ | (d) $F_{CD}$ |

**Figure 4: Comparision of Reg_EO and *FairWISA*. *FairWISA*-IRT (FairWISA-MIRT) and Reg_EO-IRT (Reg_EO-MIRT) denote *FairWISA* and Reg_EO methods implemented with IRT and MIRT as backbones, respectively. Plot (a) shows the performance in terms of accuracy, while Plots (b), (c), and (d) show the performance in terms of fairness.**

**Table 5: Comparison of F_Attr and *FairWISA* on PISA. *OECD, Gender*, and *ESCS* serve as sensitive attributes, dividing data samples into two groups for fairness evaluation. Bolding signifies the smallest value and underlining signifies the second smallest value among all methods.**

|  |  | IRT | F_OECD | F_Gender | F_ESCS | FairWISA |
|---|---|---|---|---|---|---|
| $AUC$ ↑ |  | 0.8049 | 0.8008 | 0.8029 | 0.8003 | 0.7827 |
| $ACC$ ↑ |  | 0.7316 | 0.7277 | 0.7298 | 0.7277 | 0.7130 |
| $EO$ ↓ | OECD | 0.0970 | **0.0226** | 0.0978 | 0.0847 | 0.0491 |
|  | Gender | 0.0435 | 0.0454 | **0.0081** | 0.0444 | 0.0262 |
|  | ESCS | 0.1122 | 0.1022 | 0.1124 | **0.0356** | 0.0597 |
| $NEO$ ↓ | OECD | 0.1197 | **0.0288** | 0.1199 | 0.1062 | 0.0633 |
|  | Gender | 0.0597 | 0.0623 | **0.0166** | 0.0626 | 0.0351 |
|  | ESCS | 0.1111 | 0.0965 | 0.1116 | **0.0153** | 0.0508 |
| $F_{CD \to 0}$ | OECD | 0.0498 | **-0.0317** | 0.0499 | 0.0359 | -0.0071 |
|  | Gender | 0.0235 | 0.0252 | **-0.0157** | 0.0248 | -0.0003 |
|  | ESCS | 0.0477 | 0.0348 | 0.0475 | **-0.0369** | -0.0130 |

**Table 6: Comparison of F_Attr and *FairWISA* on SLP. *Gender* and *Income* serve as sensitive attributes, dividing data samples into two groups for fairness evaluation. Bolding signifies the smallest value and underlining signifies the second smallest value among all methods.**

|  |  | IRT | F_Gender | F_Income | FairWISA |
|---|---|---|---|---|---|
| $AUC$ ↑ |  | 0.8806 | 0.8788 | 0.8791 | 0.8675 |
| $ACC$ ↑ |  | 0.8350 | 0.8329 | 0.8344 | 0.8256 |
| $EO$ ↓ | Gender | 0.0580 | **0.0329** | 0.0547 | 0.0396 |
|  | Income | 0.0417 | 0.0381 | **0.0256** | 0.0299 |
| $NEO$ ↓ | Gender | 0.1124 | **0.0386** | 0.1136 | 0.0572 |
|  | Income | 0.0713 | 0.0607 | **0.0190** | 0.0378 |
| $F_{CD \to 0}$ | Gender | 0.0241 | **-0.0142** | 0.0204 | -0.0058 |
|  | Income | 0.0146 | 0.0067 | **-0.0113** | -0.0058 |

Table 5 shows that F_OECD, F_Gender, and F_ESCS get a good fairness improvement on "OECD", "Gender", and "ESCS", respectively, and ensure a good accuracy, which suggests that the proposed $\mathcal{L}_{fair}$ as a fairness regularization is effective in improving the fairness of CD. In addition, the fairness of F_OECD, F_Gender and F_ESCS on other unknown sensitive attributes is not effectively improved, e.g., F_Gender improves the group fairness of the attribute "Gender", but still shows unfairness on "OECD" and "ESCS". Since sensitive attributes in the real world are diverse and difficult to be fully accounted for, F_Attr does not enable a truly fair cognitive diagnosis. In comparison, *FairWISA* gets a good fairness improvement on all the sensitive attributes. Similar results are also evident

in Table 6. In summary, only improving the fairness of known sensitive attributes does not yield a fair CD model, which illustrates the necessity of the study without sensitive attributes.

**Effectiveness of group inference.** To validate the effectiveness of group inference of *FairWISA* , the group labels inferred by Section 3.3 are employed for fairness method Reg_EO. Fig. 4 shows the experimental results on SLP dataset: It can be seen that both *FairWISA* and Reg_EO exhibit comparable accuracy and get a substantial improvement in the fairness metrics across backbones. This suggests that the group labels inferred by *FairWISA* are decently general and can be applied to different fairness methods. In addition, we find that on the $NEO$ metric (Fig. 4c), *FairWISA* outperforms Reg_EO. This is because Reg_EO only takes into account the consistent true positive rate of different groups, which focuses on the rate of answering questions correctly, while *FairWISA* also takes into account the fair rate of answering questions incorrectly. In conclusion, the group inference of *FairWISA* is effective and the inferred group labels are general.

**Supplementary discussion.** In addition to the above, we also discuss the relationship between inferred pseudo-labels and real sensitive attributes in Appendix C.4.1, which shows that they are not in one-to-one correspondence. Furthermore, we analyze the impact of $k$ (the number of groups in group inference) and $\alpha$ (the fairness regularization weight in Eq.(11)) in Appendix C.4.2. The results show that the performance of *FairWISA* varies regularly with these two hyperparameters.

## 5 Conclusion

In this paper, we have studied a novel and challenging problem: How to achieve fair learner modeling when sensitive attributes are unavailable. To tackle this challenge, we proposed a fair cognitive diagnosis framework named *FairWISA* that operates without relying on any known sensitive attributes. Specifically, to group learners without sensitive attributes, we introduced a pseudo-group label inference method based on leveraging unfairness maximization. To ensure fairness, we designed a fairness regularization constraint based on the principle that the model's predicted values should be similar for different groups. We theoretically demonstrated the effectiveness of *FairWISA* in improving fairness. Extensive experiments on two educational datasets have shown that our method enables achieving improved fairness without significantly degrading accuracy when sensitive attributes are unavailable.

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

## A Basic CD models used in this paper

Given the learner's proficiency vector as $h^s$, the exercise difficulty as $h^{diff}$, and the exercise discrimination as $h^{disc}$, we present the details of the three representative CD models as follows.

- **IRT** [10]. It calculates the likelihood $\hat{y}_{ue}$ of a learner $u$ answering exercise $e$ correctly using the following logistic function:

$$\hat{y}_{ue} = \frac{1}{1 + e^{h^{disc} \times (h^s - h^{diff})}}. \tag{17}$$

- **MIRT** [11]. It is an optimized version of IRT that replaces learner proficiency $h^s$ and exercise difficulty $h^{diff}$ with multidimensional vectors to accommodate more complex situations.

- **NCDM** [12]. It is a deep CD model that uses neural networks to learn complex interactions between learners and exercises. Then, NCDM predicts the likelihood of a learner $u$ answering exercise $e$ correctly:

$$\hat{y}_{ue} = MLP(Q \circ (h^s - h^{diff}) \times h^{disc}), \tag{18}$$

where $MLP(\cdot)$ is the multiple layer fully connected neural network and $Q$ is the knowledge matrix labeled by experts.

## B Pseudocode for *FairWISA*

Algorithm 1 presents the pseudocode for the implementation of *FairWISA*.

---
**Algorithm 1** *FairWISA*

---
**Input:** Set of learners $U$, set of exercises $E$, learners-exercises interaction records $R$

**Output:** A fair CD model

1: Randomly initialize the parameters $\theta$ of the CD model, group matrix $\mathcal{W}$
2: **PRE-TRAIN A BASIC CD MODEL**
3: **while** $\mathcal{L}_{CD}$ not converge **do**
4:   Sample a batch of training data $(u, e, y)$, $(u \in U, e \in E)$
5:   Compute loss $\mathcal{L}_{CD}$ by Eq. (2)
6:   Optimize the unfair CD model parameters $\theta$ by minimizing $\mathcal{L}_{CD}$
7: **end while**
8: **INFER SENSITIVE GROUP LABELS**
9: **while** $\mathcal{L}_{fair}$ not converge **do**
10:   Sample a batch of training data $(u, e, y)$, $(u \in U, e \in E)$
11:   Obtain learners' pseudo-labels according to $\mathcal{W}$
12:   Compute $\mathcal{L}_{fair}$ by Eq. (3)
13:   Update $\mathcal{W}$ to maximize $\mathcal{L}_{fair}$ with $\theta$ fixed
14: **end while**
15: **TRAIN THE FAIR CD MODEL**
16: **while** $\mathcal{L}_{total}$ not converge **do**
17:   Sample a batch of training data $(u, e, y)$, $(u \in U, e \in E)$
18:   Obtain pseudo-labels as learners' group labels according to the optimal $\mathcal{W}$
19:   Compute $\mathcal{L}_{total}$ by Eq. (11)
20:   Optimize the fair CD model parameters $\theta^*$ by minimizing $\mathcal{L}_{total}$
21: **end while**

---

**Table 7: Statistics of the datasets.**

| Dataset | #learners | #Exercises | #Records | Sensitive Attributes |
|---------|-----------|------------|----------|----------------------|
| PISA | 550,655 | 244 | 17,569,594 | Gender, OECD, ESCS |
| SLP | 3,186 | 339 | 673,471 | Gender, Family Income |

## C Experimental Supplements

### C.1 Datasets

Table 7 shows the basic information of PISA and SLP, and the detailed descriptions of them are as follows:

- **PISA** is a globally renowned assessment, with participation from approximately 100 regions or countries. It assesses the cognitive abilities of 15-year-old students (i.e. learners) in various areas such as reading and science through learner-answered exercises. Additionally, it includes questionnaires to collect sensitive information. We use the PISA2018 dataset, which focuses on the "reading" diagnostic theme. Besides, it contains three sensitive attributes, including "Gender," "OECD" (Organisation for Economic Co-operation and Development), and "ESCS" (Economic, Social, and Cultural Status). Gender is a sensitive attribute where students are divided into male and female groups. OECD is a sensitive attribute determined by nationality, where a learner's OECD=1 (or OECD=0) indicates that they are from an OECD (or non-OECD) country. OECD countries usually have better educational resources. ESCS is a sensitive attribute determined by the household wealth index, where a higher ESCS indicates a better household economy. Sthdents with a high household wealth index usually receive better education.

- **SLP** consolidates learners' academic performance data in eight subjects, including Chinese, Mathematics, English, Physics, Chemistry, Biology, History, and Geography, collected over a three-year study period through an online learning platform. Additionally, sensitive attributes regarding learners' gender, home environment, and school background are documented. SLP also contains two sensitive attributes: "Gender" and "Family Income". Family Income is a sensitive attribute based on the income level of a family. Learners with higher family incomes receive better education. Specifically, we group learners whose family income exceeds 100,000 RMB per year into one group, and the rest into another group.

For both datasets, all records are partitioned into training and test sets with an 8:2 ratio. In our experiments, the educational fairness performances of all methods is evaluated under each sensitive attribute. Please note that *FairWISA* does not utilize sensitive attribute information during the training process; instead, the sensitive attribute information is only used to assess the fairness performance during the evaluations.

### C.2 Fairness Metrics

The fairness metrics *EO* [42], *NEO* [62], and $F_{CD}$ are introduced as follows:

a) Equal Opportunity ($EO$) [42]. It requires that the model has equal true positive rates across the two sensitive subgroups.

$$EO = |P(\hat{y} = 1|y = 1, s = 0) - P(\hat{y} = 1|y = 1, s = 1)|, \quad (19)$$

where $s$ is the sensitive attribute, $y$ is the label, and $\hat{y}$ is the model's prediction. In cognitive diagnosis, it represents the difference in the proportion of correctly predicted answers by the model in the record of correct answers among different groups.

b) Negative Equal Opportunity ($NEO$) [62]. It requires that the true-negative rate is the same for s=0 samples as for s=1 samples, which is calculated as follows:

$$NEO = |P(\hat{y} = 0|y = 0, s = 0) - P(\hat{y} = 0|y = 0, s = 1)|. \quad (20)$$

$NEO$ represents the difference in the proportion of correctly predicted answers by the model in the record of incorrect answers among different groups.

c) $F_{CD}$ [18]. It requires that the CD model does not amplify discrimination in the data, which is calculated as follows:

$$F_{CD} = \left( \frac{1}{|A|} \sum_{i \in A} \hat{Y}_i - \frac{1}{|B|} \sum_{i \in B} \hat{Y}_i \right) - \left( \frac{1}{|A|} \sum_{i \in A} Y_i - \frac{1}{|B|} \sum_{i \in B} Y_i \right), \quad (21)$$

where $|A|(|B|)$ is the number of learners in group $A(B)$, $Y$ is the real correctness rate of learner, and $\hat{Y}$ is the predicted correctness rate. The closer $F_{CD}$ is to 0, the less unfair bias the CD model amplifies in the data.

### C.3 Challenging Test

To evaluate the robustness of *FairWISA*, we devise a "Challenging Test" scenario that assesses performance given distributional biases between groups. In the "Challenging Test", the fairness distribution of the data differs from the data distribution during model training. To ensure the reliability of the labels in the "Challenging Test", the interaction records are sampled from the real test data. Specifically, we change the distribution of the test set so that the correlation between the sensitive attributes and the labels is different from that in the training set. For example, in the training set, the female group may have higher correct rates, while the male group has lower correct rates. To create the challenge test data, we sampled the data so that the male and female groups in the test set had equal true correct rates. Thus, the correlation between gender and the labels no longer exists. We applied this strategy to all other sensitive attributes, thereby creating the "Challenging Test". This controlled introduction of an extreme distributional shift allows us to rigorously assess model robustness to divergent group-based patterns between training and evaluation.

### C.4 Additional Experimental Analysis

*C.4.1 Pseudo-labels and sensitive attributes.* This part analyzes the relationship between sensitive attributes and pseudo-labels inferred by *FairWISA*. In the setup, both pseudo-labels and sensitive attribute labels are binary-valued to facilitate comparison. The label accuracy in Fig. 5 is the metric to measure the correlation between these two labels. It can be seen from Fig. 5 that the accuracy of the pseudo-labels remains approximately 60% for each sensitive attribute across both the PISA and SLP datasets. It indicates that the pseudo-labels have captured sensitive attribute information to some extent. As shown in Fig. 5, the label accuracy for a single sensitive attribute is not very high, whereas the accuracy for "Mixture" is

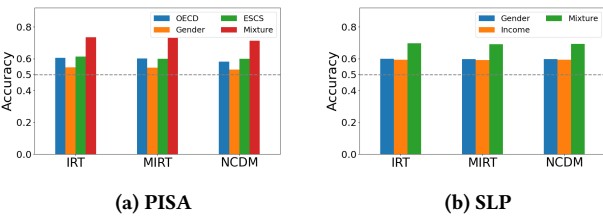

(a) PISA          (b) SLP

**Figure 5: Label accuracy of sensitive group inference. "Mixture" is the combination of multiple sensitive attributes. Take Fig. 5(b) as example, Mixture = 0 means Gender = 0 and Income = 0, while Mixture = 1 means Gender = 1 and Income = 1.**

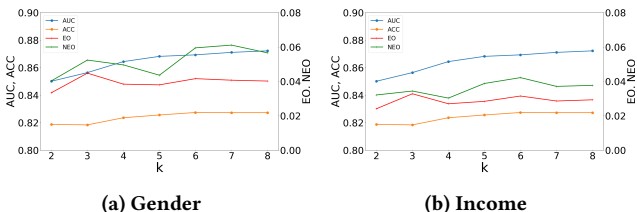

(a) Gender          (b) Income

**Figure 6: Impact of groups number $k$.**

higher. The reason is that the pseudo-labels are inferred from biases in the global data, reflecting the combined validity of all sensitive attributes rather than just a single one. For example, when the learner's attribute is "Gender=1, OECD=0", since the pseudo-label can only be either 0 or 1, it is not possible to predict both sensitive attributes correctly in this case.

Moreover, the results presented in Fig. 5 and Table 1 and Table 2 reveal that the label accuracy on a certain sensitive attribute correlates with the level of fairness impact that attribute has on the CD model. For instance, the label accuracy for "Gender" in Fig. 5(a) is the lowest, and "ESCS" is the highest. This aligns with the corresponding fairness metric values for the IRT model in Table 1, where "Gender" has the smallest value of $EO$ (0.0435) and "ESCS" has the largest value of $EO$ (0.1122). This suggests that in the PISA dataset, "ESCS" has a greater impact on the fairness bias compared to "Gender", and thus *FairWISA* learns more bias about "ESCS".

Overall, the inferred pseudo-labels exhibit some correlation with the sensitive attributes, but this correlation is not a direct one-to-one correspondence due to the inherent complexity of the learners' sensitive attributes.

*C.4.2 Parameter analysis.* We analyze the impact of the number of groups $k$ and the fairness regularization weight $\alpha$ on the performance of *FairWISA*, using IRT as the backbone for experiments on SLP.

**Impact of groups number $k$.** By varying the value of $k$ from 2 to 8, the performance of *FairWISA* in terms of accuracy and fairness of sensitive attributes ("Gender","Family Income") is observed. Fig. 6 shows that as $k$ increases, AUC, ACC, $EO$ and $NEO$ gradually increase. These observations suggest that a larger value of $k$ favors higher accuracy for *FairWISA*, albeit at the expense of fairness. This is because, with an increasing $k$, the supervisory signal for

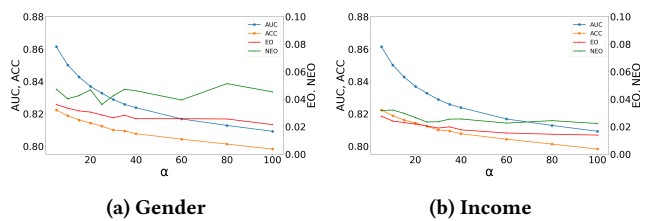

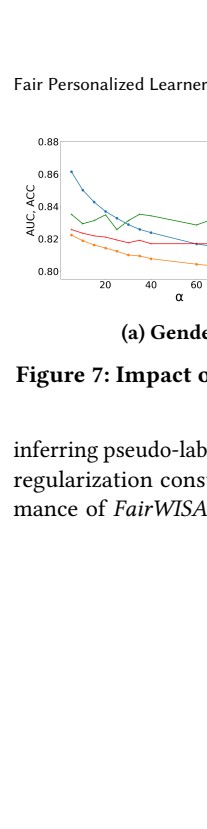

(a) Gender

(b) Income

**Figure 7: Impact of the weight of fairness regularization $\alpha$.**

inferring pseudo-labels weakens, leading to a less stringent fairness regularization constraint, thereby impacting the fairness performance of *FairWISA* . Concurrently, the accuracy is improved as *FairWISA* focuses on improving the accuracy due to the reduction of the fairness regularization weights.

**Impact of fairness weight $\alpha$**. The value of $\alpha$ is varied from 5 to 100 to observe the performance of *FairWISA* in terms of accuracy and fairness of sensitive attributes ("Gender","Family Income"). From Fig. 7, it can be seen that as $\alpha$ increases, AUC and ACC gradually decrease, and *EO* and *NEO* also have a slight decreasing trend. This indicates that as $\alpha$ increases, the fairness of *FairWISA* improves while the accuracy decreases. This is because a larger $\alpha$ increases the weight of fairness regularization, which makes the *FairWISA* focus more on fairness. Therefore, we assert that selecting an appropriate value for $\alpha$ is crucial, as it determines a balance between the accuracy and fairness performance of CD models, aligning them with the specific requirements of real-world applications.

