# OpenReview forum: "Fair Personalized Learner Modeling Without Sensitive Attributes"
_ACM.org/TheWebConf/2025/Conference — WWW 2025 Poster_

### Official Review · Reviewer_cL4b · 2024-10-30

**Novelty:** 4
**Technical Quality:** 3

**Review:**

This paper primarily investigates the issue of achieving fair learner modeling in cognitive diagnosis (CD) models. In traditional CD models, there is an issue of unfairness, where the model exhibits bias or preference towards learners with different sensitive attributes. The paper proposes a fair CD model framework, called FairWISA, which does not rely on sensitive attribute information. This framework achieves fair learner modeling by maximizing the consistency of the model’s predictions across different groups.

### Pros:

1. This work is well-organized and easy to read. The authors have done a great job in terms of writing.

2. The related work has been thoroughly reviewed.

3. The motivation behind this paper seems reasonable, provided that sensitive features are indeed unknowable in the context of CD.

4. This paper has open-sourced its implementation code.

### Cons:
1. The font of some figures in this paper is too small (e.g., Figure 4), which poses a challenge to read.

2. This work introduces some new techniques, such as a differentiable loss function, a potential sensitive information inference strategy, and theoretical guarantees for the proposed method, but in my view, the contributions of these techniques are quite limited.
Here are some reasons:

  - A differentiable fairness loss function has been proposed in previous work[1]. Please explain the advantages of the loss function proposed in this paper compared to the previous one.

  - The method proposed in this paper for inferring potential sensitive features is inefficiently high-cost because it requires training an unfair model in advance.

  - The max-min strategy proposed in this paper seems to have been introduced in previous research [2] as well. Regarding the mention in the abstract of which “tailored for CD tasks” I don’t see any specific design; it appears that it can be used for all tasks.


3. The experimental performance of this method does not outperform all the baselines, which leads me to question the effectiveness of the method proposed.




[1] Yang, Hao, et al. "Towards robust fairness-aware recommendation." Proceedings of the 17th ACM Conference on Recommender Systems. 2023.

[2] Shi, Tianhao, et al. "Fair Recommendations with Limited Sensitive Attributes: A Distributionally Robust Optimization Approach." Proceedings of the 47th International ACM SIGIR Conference on Research and Development in Information Retrieval. 2024.

**Questions:**

1. In the motivation of this paper, you claim that the privacy data and sensitive attributes are restricted by privacy concerns and legal regulations. However, your method infers potential sensitive attributes in another way. Is this approach reasonable and legal?

2. The experimental performance of this method does not outperform all the baselines. Could the author provide some reasons?

**Reviewer Confidence:**

4: The reviewer is certain that the evaluation is correct and very familiar with the relevant literature

**Scope:**

4: The work is relevant to the Web and to the track, and is of broad interest to the community

---

### Official Review · Reviewer_errU · 2024-11-07

**Novelty:** 5
**Technical Quality:** 5

**Review:**

This paper studies model-agnostic re-training techniques to improve fairness among (unknown) subgroups. Based on the empirical observation that models demonstrate similar prediction trends within the same sensitive attributes, the paper proposes to force fair training as follows. First, the proposed method clusters each data index into k different groups in order to maximize the sum of variance among predictions for positive and negative labels, respectively. Then, based on the learned clustering, the original model is retrained with the constrained objective of minimizing the original loss and the variance among identified subgroups.

---

**Technical quality**
- Overall, the **toy experiment done in the paper is sound**. The illustrative example in Figure 3 clearly demonstrates the bias in prediction, and Figure 1 also shows that forcing multiple constraints on multiple sensitive attributes may worsen the fairness among subgroups, which is interesting.
- The proposed method, which first identifies potential subgroups and then minimizes the discrepancy between subgroups, also seems **reasonable.** However, one thing **unclear to me is how the proposed method actually calculates the loss function for fairness**. In my understanding, calculating the gradient of variance and H-divergence is often difficult, and I was wondering what kind of surrogate objective is used, if there is any.
- Although not in the main text, ablation with varying number of groups and values of alpha in Appendix is also useful.

---

**Clarity**

- The paper is well-written, easy to follow.
- The motivation for enforcing fairness without labels is also adequately explained.

---

**Originality and significance.**
- Although my knowledge about related work is not comprehensive, I think ensuring fairness without providing subgroup labels or sensitive attributes is of great interest to the community. The paper presents a simple yet reasonable approach, which may be worth sharing.

---

Note for AC: I am familiar with the key concepts of fair machine learning, but not for appropriate baselines and related work.

**Questions:**

- How does the proposed method actually calculate the objective function of $\mathcal{L}_{fair}$?
- How is the hyperparameter $\alpha$ determined in the experiment? Are there also some guideline for selecting appropriate learning rates?
- What will happen when the label distribution is completely skewed across sensitive attributes, e.g., subgroup 1 has Y=1 for 90% while subgroup 2 has Y=1 for only 10%? Does the framework still work when some subgroup does not have enough data for one of the outcome labels? It may be useful to see how the result changes with the varying inherent disparities in the data.

**Reviewer Confidence:**

3: The reviewer is confident but not certain that the evaluation is correct

**Scope:**

4: The work is relevant to the Web and to the track, and is of broad interest to the community

---

### Official Review · Reviewer_zXvD · 2024-12-02

**Novelty:** 3
**Technical Quality:** 3

**Review:**

This paper suggests a fairness-aware personalized learner modeling approach for the cognitive diagnosis  task. To achieve this,  a fairness objective is first devised, and then a max-min strategy is proposed to promote the potential sensitive information inference and fair CD modeling. Additionally, some theoretical analysis is also given. Experiments on two datasets seem to validate the effectiveness of the approach.
This manuscript is well-written  and easy to follow.

**Questions:**

1. While the authors present some improvements over existing fairness-related approaches, the overall contribution of this work  seems to be somewhat incremental.
2. The motivation of this work seems weird. The authors argued existing fairness-related approaches are limited because these approaches leverage some sensitive attributes. That is, the advancement of this work is achieving fair  user modeling without using some sensitive, which is for privacy preservation. However, the contradiction is, all approaches or models were trained in a centralized manner, where all user data are collected together.  Under such a context, it is a bit unconvincing for me, maybe many users, to believe these sensitive attributes are not leaked. The authors may argue their approach can be used in a decentralized manner, but it is a bit weird to accept this motivation.
3. How about the effectiveness of this approach when some new groups of users are added whose data may be insufficient? In other words,  investigating the model fairness on some cold-start users and the model fairness on  some unseen groups is crucial.
4. Please validate the approach's effectiveness in a decentralized manner.

**Reviewer Confidence:**

4: The reviewer is certain that the evaluation is correct and very familiar with the relevant literature

**Scope:**

4: The work is relevant to the Web and to the track, and is of broad interest to the community

---

### Official Review · Reviewer_Yeqq · 2024-12-03

**Novelty:** 5
**Technical Quality:** 5

**Review:**

The paper proposes FairWISA, a fairness-aware framework for Cognitive Diagnosis models that does not rely on sensitive attribute information. Additionally, the paper introduces a fairness objective function, Lfair, to measure unfairness levels in Cognitive Diagnosis modeling.

The FairWISA framework first pre-trains an unfair (standard) Cognitive Diagnosis model and then groups learners by maximizing unfairness to derive pseudo-labels for sensitive information, using Lfair for measuring the unfairness. Finally, a fair Cognitive Diagnosis model is trained using pseudo-labels by minimizing a combined loss between cross-entropy loss (Lcd) and the fairness objective function Lfair.

**Strengths:**

- The paper highlights an important limitation of current fairness-aware Cognitive Diagnosis models, which often require access to sensitive data that is often unavailable. The motivation of the paper is well-introduced.
- The technical details behind FairWISA are sound and well-explained, and the theoretical analysis in Section 3.5 is robust.
- The experiment results show significant improvements over existing approaches.

**Weaknesses:**

- The flow of the paper could be improved in some sections. For example, there are some repetitions in the beginning of Sections 3.3 and 3.4.
- It is unclear how the observation based on Figure 3 in Section 3.2 differs from the insights from Figure 1 in Section 1.  The conclusion from the observation can be inferred from Figure 1(a), which shows that CD models consistently output larger predicted values for OECD learners. This discrepancy in predicted values intuitively results in unfair diagnosis. While this doesn't necessarily harm the core and flow of the paper, this section should be improved and made more explicit.
- Equation 9 is redundant as it is clear how  Lfair is computed from Equation 3.
- The CD abbreviation is only defined in the abstract and should be redefined in the main text.

**Questions:**

- In C.4.1, the relationship between pseudo-labels and sensitive attributes is investigated, showing a weak correlation between the two. However, the correlation is higher with "mixtures" of attributes. What is the impact of the number of groups (K) on the correlation? Was the analysis conducted with the same value of K as the number of sensitive attributes? Does the correlation increase when selecting a higher value of K, such as 2^(number of sensitive attributes)? In other words, do pseudo-labels correlate with combinations of sensitive attributes?
- An interesting observation from Figure 1 is that while CD models widen the gap between OECD and non-OECD learners, it seems to be due to an amplification of OECD learners' predictions. The predicted values of CD models for non-OECD learners do not diverge significantly from the ground truth. Rather, OECD learners' predictions seem to have been inflated. It would be interesting to further investigate this observation and examine the impact of fairness-aware models (such as FairWISA) on it.

**Reviewer Confidence:**

3: The reviewer is confident but not certain that the evaluation is correct

**Scope:**

4: The work is relevant to the Web and to the track, and is of broad interest to the community